# The Usefulness of Motor Potentials Evoked Transvertebrally at Lumbar Levels for the Evaluation of Peroneal Nerve Regeneration after Experimental Repair in Rats

**DOI:** 10.3390/jpm13030438

**Published:** 2023-02-28

**Authors:** Piotr Czarnecki, Juliusz Huber, Agnieszka Szukała, Michał Górecki, Leszek Romanowski

**Affiliations:** 1Department of Traumatology, Orthopaedics and Hand Surgery, Poznań University of Medical Sciences, 28 Czerwca 1956 r. Street, no. 135/147, 61-545 Poznań, Poland; 2Department of Pathophysiology of Locomotor Organs, Poznań University of Medical Sciences, 28 Czerwca 1956 r. Street, no. 135/147, 61-545 Poznań, Poland

**Keywords:** motor evoked potentials, lumbar transvertebral stimulation, electroneurography, rat peroneal nerve regeneration, nerve reconstruction, nerve neural transmission assessment

## Abstract

Motor evoked potentials (MEPs) are used in neurology as part of a precise diagnostic method to study the transmission of efferent neural impulses at the central and peripheral levels of the nervous system. Previous attempts have been made to apply MEPs in animal studies for evaluating neural transmission at the motor cortex center level to the muscles of the forelimbs and hindlimbs. In clinical and experimental studies, little attention is focused on the significance of the magnetic stimulation of spinal cord structures with the direct recording of the evoked potentials from peripheral nerve motor fibers. The aim of this paper was to evaluate the usefulness of the motor potentials evoked transvertebrally at lumbar levels in the evaluation of experimental peroneal nerve regeneration in rats. The bilateral transmission of efferent impulses in the distal parts of the peroneal and tibial nerves was verified by recordings of evoked potentials following transvertebral magnetic stimulation at lumbar levels (MEPs) and the electrical stimulation of the sciatic nerve in classical electroneurographic (ENG) tests for comparison. Recordings were performed 24 weeks after grafts on surgically treated hindlimbs as well as on non-operated hindlimbs as controls. Both the MEP and ENG stimulations resulted in evoked potentials with larger amplitude values following the application of the magnetic pulses, with more being recorded on the non-operated hindlimbs than on the operated ones when recordings were taken from peroneal nerve branches. We observed statistically significant correlations between the MEP and ENG results for peroneal and tibial nerve amplitude on the non-operated side and peroneal nerve amplitude on the operated side. The recorded latencies of the evoked potentials were shorter in the ENG studies than in the MEPs for the non-operated side. The results demonstrated the phenomenon of regeneration in the motor fibers of the peroneal nerves 24 weeks after grafting in the experimental conditions. In this study, the MEPs were as useful as the ENG studies for evaluating regeneration in the motor fibers of hindlimb nerves in rats, although they were not significantly different. This paper discusses the clinical importance of transvertebral MEPs induced at the lumbosacral and cervical levels with a magnetic field for the diagnostic evaluation of efferent impulse transmission at different levels of the motor pathway.

## 1. Introduction

Motor evoked potentials (MEPs) are used in neurology as part of a diagnostic method to study efferent transmission at both the central and peripheral levels of the nervous system. The extension of this technique with a transcranial application of stimuli, called rTMS, is widely used in the treatment of neurological and psychiatric disorders such as Alzheimer’s disease, Parkinson’s disease, epilepsy, stroke, depression and anxiety disorders, and sleep disorders [1,2,3,4,5,6].

Currently, little attention is focused on the magnetic stimulation of spinal cord structures with the direct recording of evoked potentials from peripheral nerve motor fibers. Although many methods are used for the assessment of nerve regeneration in rat models, there is little information about transcranial magnetic stimulation (TMS) or transvertebrally induced spinal magnetic stimulation and motor evoked potentials (MEPs) [7,8,9,10,11,12,13].

Transvertebral magnetic stimulation is casuistically described in the literature, and some similar techniques have been proposed [12,14,15,16]. Kagan et al. used a handcrafted solenoid coil to stimulate the sciatic nerve directly. Their work demonstrated that the smallest coil must have an outer diameter of at least 5 mm, as smaller ones do not stimulate the nerve [17]. Boonzaier et al., in their work on rats, compared the effects of using two coils of different diameters. One was the standard coil with a diameter of 50 mm (the same as that used in our work here), and the other was miniaturized, dedicated to rodents, with a diameter of 25 mm. Their simulations predicted that the large conventional TMS coil would not achieve lateralized focal stimulation of the rat motor cortex. They hypothesized that the small rodent-specific TMS coil would elicit more focally lateralized MEPs than the large coil. However, unexpectedly, the in vivo data showed that both the TMS coils were able to induce focal stimulation and elicited MEPs from the rat contralateral forelimb. Coil size reduction is still extremely challenging because of the heating and mechanical stress generated from electromagnetic forces due to the increased resistance and larger currents needed to produce an effective magnetic field [18]. Silicone oil cooling was applied in a small rodent-specific coil that could deliver focal stimulation pulses at intensities similar to conventional TMS coils [14]. Transvertebral magnetic stimulation can eliminate some of the problems associated with direct nerve stimulation that authors report, such as the proximity of stimulating and recording electrodes to each other, stretching tested nerves, variable ambient temperature, the dehydration of tissues, and difficulties with the correct placement of the ground electrode [19,20]. During our study described here, special attention was paid to ensuring that the abovementioned factors did not affect the course of the experiment, namely, the location of the grounding electrode, insulation, and the lack of stretching of the nerve at the point of application of the recording and stimulating electrodes were checked, and the tissues were moistened with saline. Some authors state that, during MEP applications, the spinal white matter fibers of efferent pathways are not sufficiently sensitive to allow selective excitation or that they are difficult to selectively excite. It is assumed that the application of MEPs to the motor fibers in ventral roots with muscle recordings is considered in clinical neurophysiology to be the only acceptable method of spinal excitation [7,12,14,15].

The aim of this study was to evaluate the usefulness of motor potentials evoked transvertebrally at lumbar levels in the evaluation of experimental peroneal nerve regeneration in rats after end-to-side coaptation. Three different surgical techniques were used to graft the peroneal nerve to the tibial nerve, including grafting without an epidural window, with an epidural window, and with a 180-degree free peroneal nerve graft rotation. Bilateral transmission of efferent impulses in the distal parts of peroneal and tibial nerves was verified by recordings of the evoked potentials following transvertebral magnetic stimulation at lumbar levels (MEPs) and the electrical stimulation of the sciatic nerve in classical electroneurographic tests (ENG) for comparison.

## 2. Materials and Methods

### 2.1. Animals

Forty-five female Wistar rats weighing from 197 to 260 g (average 229 g ± 14 g standard deviation (SD) were used in the experiments. Ethical considerations were in accordance with the Helsinki Declaration, and approval was also received from the Bioethical Committee of the University of Medical Sciences in Poznań, Poland (decision No. 1279/18). Animals were intraperitoneally anesthetized with ketamine (Ketanest) at a dose of 90 mg/kg of body weight. The effect of the drug started about 20 min after administration and lasted for about 2 h. In addition, we injected the operated area with lignocaine solution at a concentration of 1%. Attempts to measure the blood pressure and PCO_2_ in the external carotid vein using a small cannula were abandoned because they influenced the arterial flow, an important factor influencing neurophysiological parameters. To prevent the aspiration of saliva into the lungs, the animals were premedicated with atropine sulfate (*Polpharma*) at a dose of 0.05 mg/kg. The study group included data from peroneal nerve reconstruction using three techniques: autologous nerve grafting (n = 15) and end-to-side neurorrhaphy to a tibial nerve with or without an epineural window (n = 15 for each). Neurophysiological studies were performed 24 weeks after surgery. The control group consisted of data from ENG and MEP examinations of the corresponding nerve trunks on the opposite side (non-operated). This partially eliminated individual variations in the measured values. 

### 2.2. Surgical Procedures

Microsurgical and atraumatic techniques were applied using appropriate tools and an appropriate operating microscope magnification. The incision into the skin was made along the edge of the iliac crest, gluteal muscles were detached, and the space where the sciatic nerve divides into the peroneal, tibial, and sural nerves was revealed.

Then, the three types of peroneal nerve reconstruction described above were performed. The minimum number of Ethilon 10-0 (Ethicon) non-absorbable monofilament sutures were used.

The tibial nerve was used as a donor in the peroneal nerve reconstruction. This choice was made considering the technical aspect of sewing a thinner peroneal nerve to a tibial nerve of twice the diameter. The tibial nerve also includes almost twice as many nerve fibers and should, theoretically, facilitate a greater number of potentials in donor regeneration as reflected in clinical care (Figure 1).

### 2.3. Neurophysiological Testing

Neurophysiological examinations included recordings of the electroneurographic (ENG) stimulation studies of the hindlimb motor fibers and evaluation of the total efferent neural transmission studies from lumbar spinal centers to the distal parts of nerves and effectors (MEPs) (Figure 2). The tests were conducted in the same room at 21–23 °C. Recordings and stimulations were performed using the KeyPoint Diagnostic System (Medtronic, A/S, Skovlunde, Denmark). This study used minimally invasive magnetic and electrical stimuli, which means that their strength was adjusted to elicit MEPs and ENG potentials with supramaximal amplitudes while minimizing movement artifacts for the stimulated objects that could affect the recording conditions.

Standard single pulses of the magnetic field were used for transvertebral stimulation to induce motor evoked potentials (MEPs). They were induced using a 50 mm diameter circular coil placed bilaterally over the descending fibers of white matter at the L3-L5 spinal cord level from a MagPro R30 (Medtronic A/S, Skovlunde, Denmark). Recordings were performed using the MEP technique 10 mm from the peripheral graft. The optimal site for stimulation was defined with tracking stimuli delivered at 1 Hz from 5–60% of the maximal stimulus strength (1.5 Tesla), while the maximal amplitude of MEPs was recorded from the nerve. A pair of silver chloride hook-shaped recording electrodes was used. The anode of the electrode pair was oriented closer to the spinal center, while the cathode was oriented distally. The distance between the poles of the recording electrodes was 3–4 mm. The ground electrode was placed on the muscle close to the recording electrodes. During the recordings, special care was taken to avoid drying the dissected nerve branches; they were soaked with drops of warm paraffin oil. The highest intensity stimulus never exceeded 60% of the maximal stimulus output. The recordings were performed at an amplification of 50–5000 µV and a time base of 2–5 ms. All the MEP recordings were performed using 0.5 Hz low-pass filter settings, while the upper-pass filter of KeyPoint was set to 2 kHz. The outcome measures were the amplitudes (in µV) and latencies (in ms) of the MEPs.

Electroneurography of the sciatic nerves was used to bilaterally detect the changes in the transmission of neural impulses following the surgical nerve grafts. Following the application of electrical, rectangular pulses with a duration of 0.2 ms at 1 Hz and an intensity from 0 to 40 mA delivered from the bipolar stimulating silver electrodes in the proximal part of the sciatic nerve, M-waves were recorded from the distal parts of the peroneal and tibial nerves with other pairs of bipolar silver electrodes. Recordings of these potentials verified the transmission of neuronal impulses in the peripheral motor fibers. The recordings were performed at an amplification of 5–5000 µV and a time base of 2–10 ms. The outcome measures were the amplitudes (in µV) and latencies (in ms) in the M-wave potential recordings. Donor tibial and graft peroneal nerve fibers were excited following electrical stimulations of sciatic nerve rectangular pulses with a duration of 0.1 ms at 1 Hz and a strength from 0.06 to 1 mA (milliamperes) delivered from the KeyPoint stimulator. Bipolar silver hook electrodes stimulated the sciatic nerve and recorded the output from the peroneal and tibial nerves. A distance of 3 mm between the anode and cathode was maintained. A ground electrode was placed on the semitendinosus muscles. 

### 2.4. Statistical Analysis

Collected data were statistically analyzed using the StatSoft Statistica 13.3 software. Descriptive statistics included minimal and maximal values (range), mean, and standard deviations (SD) for measurable values. The cumulative data from the three investigated groups were used to calculate the latency and amplitude. The amplitude and latency results for both MEPs and ENGs were compared using Student’s *t*-test for independent and dependent groups. Spearman’s rank correlation was used to assess the relationship between the amplitude and latency results for MEPs and ENGs and evaluate the MEP accuracy according to typical ENG studies. *p*-values of less than 0.05 were considered statistically significant.

## 3. Results

Comparing the results obtained with the MEP and ENG methods showed no statistically significant differences between the amplitudes of the peroneal and tibial nerves on the non-operated side. The same was observed for the tibial nerve on the operated side. However, a significantly higher amplitude was observed in the MEP recordings for the peroneal nerve on the operated side (Figure 2). The average values, *p*-values of the side comparison, and Spearman’s correlation parameters between the MEP and ENG methods are presented in Table 1. 

The latency parameter differed significantly between the methods in all the comparisons, depending on the distance from the stimulation to the recording site. 

Spearman’s correlations were significantly positive between the MEP and ENG results for the peroneal and tibial nerve amplitude parameters on the non-operated side and the peroneal nerve amplitude on the operated side. The amplitudes recorded with the MEP and ENG methods in the tibial nerve on the operated side were not significantly correlated. 

Statistically significant correlations were found between the MEP and ENG results for peroneal nerve latency on the operated side, but they were not correlated for the non-operated side. The results for tibial nerve latency were not statistically correlated on both the operated and non-operated sides when comparing direct electrical and transvertebral magnetic stimulation.

## 4. Discussion

The presented study is part of a wider range of experimental studies evaluating differences in motor fiber regeneration when comparing three methods of nerve repair. In this paper, we aggregated the results to conduct a more robust statistical analysis and compared the two magnetic and electrical stimulation methods.

We used a coil with the smallest available diameter possible for small animals to generate a stimulating magnetic beam (60% of the maximal strength) that was less than 5 mm and able to penetrate tissues 3 cm deep in a straight line from the superficial plane until spreading according to the laws of magnetic fields. Using a smaller coil (dedicated to small animals), such as that in the studies of Kagan et al. [17], would probably improve the method’s accuracy by eliminating interference with the recording electrode and providing a more precise stimulation point. In contrast, transvertebral magnetic stimulation, as a minimally invasive technique, satisfactorily and sufficiently penetrated the soft tissue structures and spine bones to reach the bundles of fibers as well as the cell bodies of motor neurons. It should be assumed that, in this case, the expected effect was due to the use of a coil that was 5 mm in diameter. 

The greatest contributor to false positive readings in direct stimulation is interference from the large amplitude stimulus artifact recorded in ENG studies when the stimulating and recording electrodes are close to each other or there is interference with muscle impulses due to tissue contraction. This phenomenon was described by Rupp et al. [19] in their studies on rats. They excised a 14 mm segment of the sciatic nerve and ligated the proximal stump to prevent regeneration. The injury to the sciatic nerve was performed 4 mm before the submersion of the branch into the gastrocnemius muscle. They then directly stimulated the proximal sciatic nerve stump at different levels. Despite the lack of continuity of the sciatic nerve to the gastrocnemius muscle, an electrical response was obtained from this muscle. The more proximal the nerve stimulation and the greater the amount of muscle group stimulation obtained, the more significant the response from the gastrocnemius muscle. To confirm the phenomenon of tissue impulse conduction, the authors gradually resected the individual muscle groups innervated by the sciatic nerve. This resulted in diminishing electrical readings from the denervated gastrocnemius muscle following the stimulation of the proximal sciatic nerve stump [19].

On the other hand, the MEP test is also associated with some difficulties due to its high sensitivity to the coil orientation, the distance from the spinal cord, and, most of all, the motor artifacts from the several muscle groups stimulated [3,12,13].

In their study, Kegan et al. assessed the variability of magnetic stimulation in animals by conducting research on rats. They compiled ENG recruitment curves from all the animals (8) in which curves containing the full range of ENG responses were generated using the same coil and the same range of stimulation intensities. They observed that only modest changes in stimulus intensity were needed to evoke the required responses across the animals. These results indicate that magnetic stimulation can reliably and reproducibly induce activation [17], similarly to our study. 

One of the variables that should be considered and that may affect the results of nerve conduction studies is the inter-animal variability resulting from physiological differences between animals. Our study partially reduced the risk of this variable affecting the results. The data were the results of electroneurographic examinations of the respective nerve trunks in each rat—for the study group, this was the lower limb with nerve reconstruction. In contrast, for the control group, it was the opposite lower limb without surgery. This partially eliminated individual variation in the measured values. Another limitation of the presented studies was the quality of the ENG recordings, which can be affected by high-amplitude electrical stimulus artifacts that sometimes interfere with the onset of the basal evoked potential. A possible solution to this problem is to use equal numbers of positive and negative stimuli that add up to zero when the sum of the action potential is generated. However, this requires the creation of a special software algorithm for the stimulation and the recording of short-latency ENG potentials, which are induced over a short conduction distance between the stimulation and recording electrodes. The methodology’s weakness in terms of recordings of nerve conduction velocities in animals, especially small ones, is due to the inaccuracy of the conduction distance measurements between the pairs of stimulating electrodes releasing the electrical pulse and a pair of recording electrodes. This issue was mentioned by other authors of previous experimental studies [20,21]. The same holds true for measuring the conduction distance between the stimulating coil releasing the magnetic field transvertebrally and a pair of recording electrodes from nerve stumps or muscles [22]. Therefore, we did not analyze this parameter in detail, as it would not provide additional information for comparison. If in the future nerve conduction values need to be compared, magnetic stimulation appears to be an inferior method compared to direct electrical stimulation.

Human clinical studies on using MEPs for transvertebral magnetic stimulation show marked differences between normal subjects and subjects with disc–root conflict in preoperative diagnostics. One of the essential advantages of indirect nerve conduction assessment with MEPs is the possibility to conduct the examination non-invasively compared with the direct evaluation of nerve conduction with ENG, which is often performed intraoperatively [16,18,19]. When MEPs are recorded in elderly subjects with advanced muscle atrophy caused by axonal pathological changes, especially in those with lumbosacral chronic disc–root conflicts, this method may show limitations because fewer muscle motor units react to the excitation applied with the magnetic stimuli. Significant neurogenic changes in muscles, with the first visible signs of muscle mass reduction caused by sarcopenia in “healthy subjects” aged 50 years and more, which are usually not the result of degenerative changes in motor axons, have been reported [16]. This phenomenon may better explain the results of electroneurographic studies than electromyographic recordings for the same patient. Perhaps MEPs recorded from nerves with surface electrodes along their anatomical passages are more stable than those recorded from muscles for the diagnostic purposes of clinical neurophysiology. They can be characterized by parameters that are different from those of MEPs recorded from muscles due to nerve excitation properties. Testing this assumption will require additional comparative studies in healthy subjects and patients with lumbosacral disc–root conflicts. 

Considering the further clinical significance of MEPs applied transvertebrally at the cervical level for the diagnosis of brachial plexus function, together with other neurophysiological techniques, such as electroneurography, needle electromyography, and measuring somatosensory evoked potentials, it is possible to assess the proximal part of the peripheral motor pathway, between the cervical root level and Erb’s point, and via trunks of the brachial plexus to the target muscles. This may be of particular importance in the case of damage to the brachial plexus in its proximal part [23]. The available data from the literature on the use of magnetic stimulation in the assessment of the proximal nerve part of the upper extremity mostly relate to neurological disorders such as motor neuron disease, neurogenic thoracic outlet syndrome, chronic inflammatory demyelinating polyneuropathy, and Guillain–Barré–Strohl syndrome [24]. 

Transvertebral magnetic stimulation enables the recording of the supramaximal potential, which results from the stimulation of the entire axonal pool of the tested motor path, similar to testing with an electric stimulus. Two types of stimulation, magnetic and electrical, can be selected for one examination, depending on the individual patient’s diagnostic protocol, and the evoked potentials’ parameters can be compared. It is commonly believed in clinical neurophysiology that, when the results of ENG examinations of peripheral nerve conduction and MEPs verify the total efferent conduction from the motor center to the muscles, are correct, invasive needle examinations may be abandoned in the diagnostic process. This is especially important in the case of pediatric patients.

MEP probing is a valuable addition to classical direct stimulation, and in our results, significant correlations were found between the two tested methods when amplitude measurements were considered. A possible explanation for the significantly higher values of the amplitudes in the MEP rather than in the ENG recordings of the operated peroneal nerves may be the more generalized effect of the magnetic stimuli exciting both the spinal motor centers and the ventral roots rather than electrical stimuli with silver-wire electrodes involving most parts but not the whole nerve stump. Latency results are less reliable due to the short distance between the stimulation and recording sites. The well-known parameter in the literature of the conduction velocity was also not analyzed in our study in detail because the distance between the stimulation and recording points was constant, and such analysis would not have provided additional information.

## 5. Conclusions

In this study, we found that the amplitude parameters of the transvertebrally induced MEPs were very similar to those recorded from nerves following electrical stimulation. In interpreting the MEP recordings for the transmission of the motor pathway, there was no interference from artifacts resulting from an electrical stimulus’s action, which occurs in ENG stimulation tests.

In experimental studies, when confirmation of the regeneration process in the peripheral nervous system is expected, nerve MEP recording can be successfully implemented to evaluate nerve impulse transmission within axons of the whole efferent pathway from the spinal motor center. MEPs are as useful as ENG studies for evaluating regeneration in the motor fibers of hindlimb nerves but do not demonstrate any evident superiority. The clinical importance of MEPs induced transvertebrally at the lumbosacral and cervical levels by a magnetic field in the diagnostic evaluation of efferent impulse transmission at the different levels of the motor pathway is crucial for the personalized diagnosis of patients with motor disorders of different levels of advancement. 

## Figures and Tables

**Figure 1 jpm-13-00438-f001:**
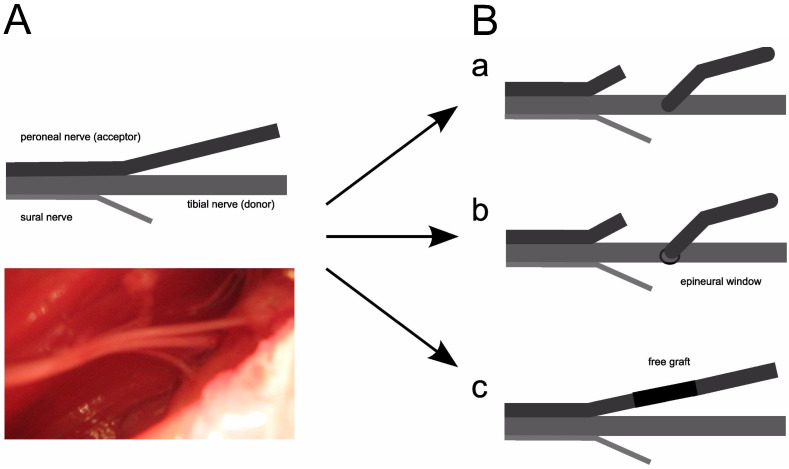
(**A**) Scheme of the anatomical division of the sciatic nerve into three branches (schematic drawing and its anatomical microphotograph). (**B**) Three techniques of surgical procedures that were performed (a—peroneal nerve graft to a tibial nerve without window incision, b—the same procedure with window incision, c—free graft of peroneal nerve using same nerve when its part was rotated 180° and sewed again).

**Figure 2 jpm-13-00438-f002:**
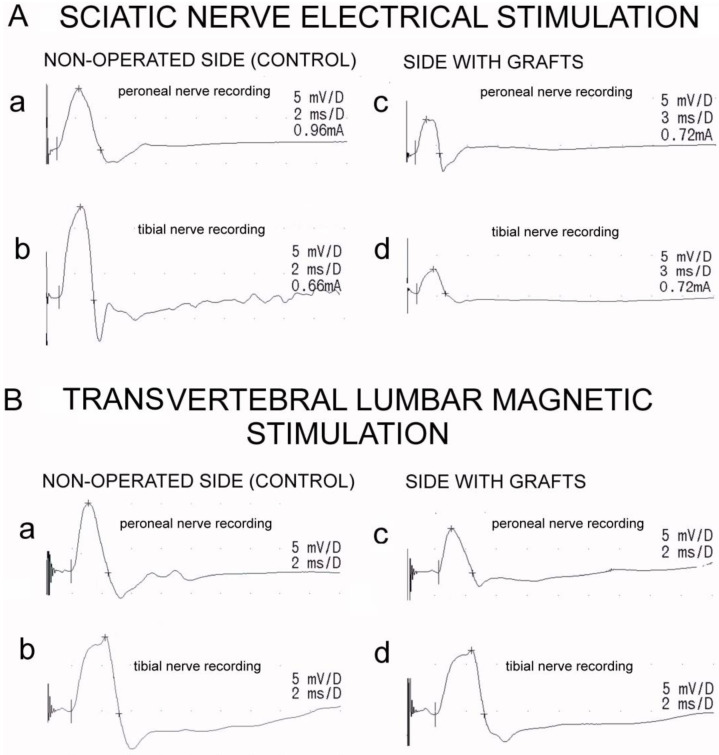
Examples of motor potentials evoked electrically (**A**) and (magnetically) in one rat from the group with the epineural window graft. Recordings were performed from peroneal (a) and tibial (**B**) nerves on the non-operated side as well as from peroneal (c) and tibial (d) nerves on the side with grafts.

**Table 1 jpm-13-00438-t001:** Comparison of the results from neurophysiological tests in three groups of rats (cumulative data). The mean values with standard deviations are presented.

Type of Test Recording Site		Measured Parameter	Non-Operated Side Control	Operated Side	Non-Operated vs. Operated *p*	r_s_	*p*
**ENG**	Peroneal nerve	Amplitude (µV)	8476 ± 5798	6782 ± 3492	0.426689	0.54	**0.001**
		Latency (ms)	0.93 ± 0.26	0.96 ± 0.25	**0.000001**	0.11	0.482
	Tibial nerve	Amplitude (µV)	9907 ± 6084	5898 ± 4317	0.921025	0.46	**0.005**
		Latency (ms)	0.94 ± 0.35	1.12 ± 0.39	**0.000001**	0.14	0.391
**MEP**	Peroneal nerve	Amplitude (µV)	9605 ± 6622	6970 ± 3492	**0.019148**	0.41	**0.010**
		Latency (ms)	1.62 ± 0.28	1.54 ± 0.45	**0.000001**	0.40	**0.008**
	Tibial nerve	Amplitude (µV)	9767 ± 5996	5898 ± 4317	0.238960	0.13	0.428
		Latency (ms)	1.65 ± 0.33	1.12 ± 0.39	**0.000001**	−0.09	0.585

ENG—electroneurography; MEP—motor evoked potential; r_s_—Spearman’s rank correlation of the test results; *p*-values ≤ 0.05 are considered statistically significant for comparisons of mean values and for rank correlations; bold letters indicate the statistically significant differences.

## Data Availability

The datasets analyzed for this study can be found in the repository of the first author.

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
