# Peer review of "The Usefulness of Motor Potentials Evoked Transvertebrally at Lumbar Levels for the Evaluation of Peroneal Nerve Regeneration after Experimental Repair in Rats"

_jpm, 2023, doi:10.3390/jpm13030438_

Round 1

Author Response

Dear Reviewer,
We greatly appreciate your valuable comments regarding our manuscript, enabling improving the quality of our work. We appreciate and accept all remarks and suggestions. Please find the responses point by point in the in attached file.

Sincerely yours,
Authors

Reviewer 2 Report

The authors present an experimental study in rats where nerve regeneration Is evaluated using a non-invasive method of generating motor evoked potentials, namely through magnetic stimulation. 

According to the authors line of argumentation, the main disadvantage of direct electrical stimulation is an interference phenomenon between stimulation and recording electrode as well as the imminent risk of neural trauma at the stimulation site. 

The experimental setting is a suture/grafting model of the sciatic nerve in rats, the unoperated side serving as control. Three experimental groups are established: End to side coaptation without epineural incision, end to side coaptation with epineural incision and nerve grafting. Data evaluation is cumulative, comparing the results of MEP vs ENG stimulation.

As far as I understand the results section, there are no significant differences between the two methods, when examining an unoperated nerve. Monitoring of latency gave no meaningful results due to the inaccuracy of length measurement. There were significant higher values for amplitude in the operated peroneal nerve in the MEP group as compared to the ENG group. 

 1.    The term for the surgical restoration of a peripheral nerve, suture or graft, is Coaptation (as opposed to anastomosis, which refers to the restoration of the continuity of a vascular structure). 

2.    Certain parts of the discussion should be transferred to the introduction, e.g. lines 187 to 193, as well as the considerations on the physics of coil sizes and shapes (lines 161 – 186).

3.    Stimulation artefacts in ENG analysis can be eliminated by applying an equal number of positive and negative stimuli, which will add to zero when the sum action potential is generated.

4.    What is the point setting up three surgical groups, when the data evaluation is performed in cumulo? (The underlying study design with three different methods of nerve coaptation should reveal differences in the quality of axonal regeneration, resulting in difference of amplitude measurements)

5.    The authors do not discuss the statistical results, e.g. why is the amplitude in operated peroneal nerves significantly higher in MEP compared to ENG.

6.    The authors use a 5 cm coil on a rat with an average body length (excluding the tail) of less than 20 cm. I have difficulties to understand how a precise point of stimulation can be generated, as well as precise measurement of the distance between the point of stimulation and the point of recording. The authors seem to be aware of this shortcoming (lines 164, 236)

7.    What do the authors mean by minimally invasive MEP stimulation? This should be clarified in the Materials and Methods section.

8.    What do the authors mean by a beam of 5 nm (Line 163)

9.    Line 102 – 10 cm?

10. I  do not agree with the conclusion of the authors stating that MEP analysis is a precise method for the evaluation of regeneration in peripheral nerves. In contrast, he data suggest, that basic information about ongoing regeneration can be obtained with MEP, however, the superiority of this method in comparison to ENG is not evident.

11. A professional translation of the article is recommended.

Author Response

(The authors gave the same response as above.)

Round 2

Reviewer 2 Report

The requested changes have been made by the authors. However, the issue of translation remains. A fellow researcher, who is British, was so kind to check the text for me. He states that "several parts of the manuscript are poorly written, making it difficult to get a clear understanding of the intended meaning". He is afraid, that "most readers will find the manuscript difficult to read and confusing". This is very much to the point and reflects what I felt when first reading the text. I am sure a professional translator will make ease of dealing with this matter. Along this line, my colleague also mentioned a certain weakness in terminology, the most prominent example being "oververtebrally" which appears in the title as well as in the text. A suggestion would be the use of "transvertebral" stimulation, in analogy to the more widely used term "transcranial stimulation".

Finally, bearing in mind that the article under review is part of a larger project of research, it is important to state the weakness of the methodology for the recording nerve conduction velocities, due to the inaccuracy of length measurements with (in comparison to the size of the animal) comparatively large coils sizes. If in future nerve conduction values are to be compared, magnetic stimulation appears to be the inferior method compared to direct electrical stimulation (unless proved otherwise in an experimental setting), and this should be addressed in the discussion of this article as well as considered when setting up the experimental design for future experiments. Citations in this context will certainly be necessary, I did not mention this in my first review, but would like to raise the point now
